# Concordance of randomised controlled trials for artificial intelligence interventions with the CONSORT-AI reporting guidelines

Alexander P. L. Martindale [1], Carrie D. Llewellyn[2], Richard O. de Visser[2], Benjamin Ng [3,4], Victoria Ngai [5], Aditya U. Kale[6,7,8], Lavinia Ferrante di Ruffano [9], Robert M. Golub[10], Gary S. Collins [11], David Moher [12], Melissa D. McCradden [13,14,15], Lauren Oakden-Rayner[16], Samantha Cruz Rivera[17,18], Melanie Calvert [8,17,18,19,20], Christopher J. Kelly[21], Cecilia S. Lee [22], Christopher Yau[23,24], An-Wen Chan[25], Pearse A. Keane [26], Andrew L. Beam [27,28], Alastair K. Denniston [6,7,8,17,26] & Xiaoxuan Liu [6,7,17] ✉

The Consolidated Standards of Reporting Trials extension for Artificial Intelligence interventions (CONSORT-AI) was published in September 2020. Since its publication, several randomised controlled trials (RCTs) of AI interventions have been published but their completeness and transparency of reporting is unknown. This systematic review assesses the completeness of reporting of AI RCTs following publication of CONSORT-AI and provides a comprehensive summary of RCTs published in recent years. 65 RCTs were identified, mostly conducted in China (37%) and USA (18%). Median concordance with CONSORT-AI reporting was 90% (IQR 77–94%), although only 10 RCTs explicitly reported its use. Several items were consistently under-reported, including algorithm version, accessibility of the AI intervention or code, and references to a study protocol. Only 3 of 52 included journals explicitly endorsed or mandated CONSORT-AI. Despite a generally high concordance amongst recent AI RCTs, some AI-specific considerations remain systematically poorly reported. Further encouragement of CONSORT-AI adoption by journals and funders may enable more complete adoption of the full CONSORT-AI guidelines.

Artificial intelligence (AI) has been introduced to healthcare with the promise of assisting or automating tasks to reduce human workload. In publications, medical AI models have been reported to produce promising results in a variety of data-driven scenarios, including clinical decision support, medical image interpretation and risk prediction[1–3]. However, real-world implementation of medical AI interventions has so far been limited and the potential benefits not yet realised. One significant barrier to adoption is the lack of high-quality evidence supporting their effectiveness, such as from randomised controlled trials (RCTs) performed in relevant clinical settings[4,5].

RCTs provide the highest quality evidence for evaluating the impact of medical interventions. Importantly, they provide evidence on the effect of interventions on outcomes grounded in benefit to patients and the health system and often generate sufficient evidence to justify widespread adoption. Therefore, it is imperative that RCTs are well-designed, properly conducted and transparently reported. Incomplete or unclear reporting results in poor transparency of bias and research waste, leading to poor decision-making and non-reproducibility of findings[6].

Reporting guidelines such as the CONSORT 2010 statement set out consensus-driven minimum reporting standards for the reporting

**Fig. 1 | PRISMA flow diagram[74].**

of RCTs[7]. To provide additional and specific guidance for RCTs involving AI interventions, the CONSORT-AI extension was developed and published in September 2020[8]. CONSORT-AI includes 14 additional checklist items to be reported alongside the 37 CONSORT 2010 items. These items provide elaboration and additional criteria specific to AI, such as reporting algorithm version and input data selection, aiming to improve the completeness and relevance of the original CONSORT statement to AI interventions[8].

Many RCTs of AI interventions have been published since CONSORT-AI, but the completeness of reporting is currently unclear. This systematic review aims to assess the completeness of reporting in recent RCTs for AI interventions using CONSORT-AI and to summarise study characteristics to provide insight into this area of research.

## Results

In total, 5111 articles were retrieved following deduplication. 332 articles were selected for full-text review following title and abstract screening. 267 articles that did not meet the inclusion criteria were excluded, including 104 ongoing or unpublished trial registry entries. 65 RCTs met the inclusion criteria and were included in the final analysis[9–73]. Amongst these, four were RCTs of diagnostic test evaluation, where the primary outcome was diagnostic yield (for example, the effect of an assistive AI intervention on a clinician's ability to detect disease)[13,17,24,36]. Whilst these interventional studies did not measure patient outcomes, they were included in this review as the concordance with CONSORT-AI guidelines remains relevant. Details of excluded articles are shown in the PRISMA flow diagram[74], see Fig. 1. The full list of included RCTs is available in Supplementary Data 1.

## Study characteristics

The majority of studies were conducted in China ($n = 24$, 37%)[12,15,27,29,30,37–41,43,59–66,68,69,71–73], USA ($n = 12$, 18%)[16,19,22,25,28,32,42,51,55,56,67,70] and Japan ($n = 5$, 8%)[9,24,31,33,50]. There were 4 international multicentre studies conducted across European sites[11,13,34,44] and 10 studies performed within individual European countries: France ($n = 3$)[18,36,58], Italy ($n = 2$)[53,54], Spain ($n = 2$)[21,47], England ($n = 1$)[26], Germany ($n = 1$)[48] and Denmark ($n = 1$)[14]. The remainder ($n = 10$, 15%) took place across South Korea, Taiwan, India, Thailand, Israel, Mexico, Argentina, Rwanda and Malawi, as shown in Fig. 2[10,17,20,23,35,45,46,49,52,57].

Median sample size across all included RCTs was 186 (IQR 56-654). Most RCTs were single centre ($n = 39$, 60%) versus multicentre ($n = 26$, 40%). Studies were commonly unblinded ($n = 24$, 37%) or single-blinded ($n = 21$, 32%), with few double-blinded RCTs ($n = 2$, 3%). 18 (28%) did not report details of any blinding.

## Types of AI intervention

The most common types of AI intervention were endoscopy assistance ($n = 13$, 20%), image enhancement ($n = 11$, 17%), image classification ($n = 9$, 14%), and chatbots ($n = 7$, 11%). Endoscopy assistance was defined as computer-aided detection of suspicious lesions during colonoscopy or upper endoscopy, which highlight regions on the endoscopist's display in real-time. Image enhancement encompasses AI interventions that modify medical images, such as ultrasound or radiography, to improve clarity or highlight areas of interest. In contrast, image classification involves automated diagnosis or interpretation of medical images using an AI, with the results informing clinician decision-making. Chatbots use language models to process

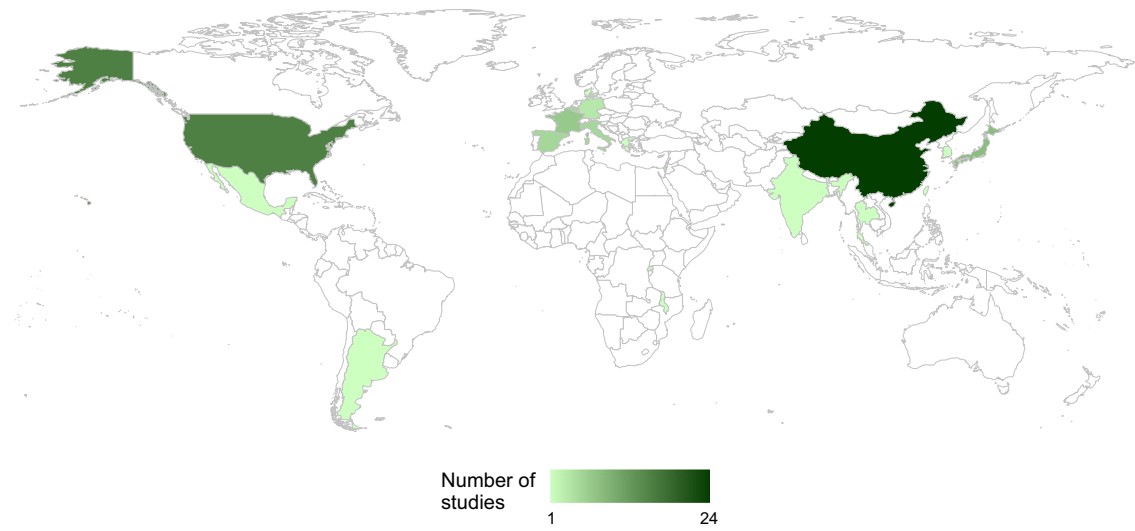

**Fig. 2 | Location heatmap of included studies by country, showing high distribution within China and USA.** Generated using R Statistical Software (v4.1.1, R Core Team 2021).

### Table 1 | Type, frequency and description of AI interventions across included studies

| Classification of intervention | Frequency | Description of classification | Subclassification |
|---|---|---|---|
| Endoscopy assistance | 13 | Computer-aided detection of suspicious lesions during endoscopy procedures, usually with an image overlay. | Colonoscopy ($n = 10$), Upper endoscopy ($n = 3$) |
| Image enhancement | 11 | Modification of medical images to enhance clarity or highlight areas of interest to guide clinicians (excluding endoscopy). | Clarity enhancement ($n = 7$), Image overlay ($n = 4$) |
| Image classification | 9 | Automated diagnosis or interpretation based on images. | Standard photograph ($n = 3$), Radiographs ($n = 2$), Fundus imaging ($n = 2$), Echocardiography ($n = 1$), Chest CT ($n = 1$) |
| Chatbots | 7 | Use of natural language processing to interact with humans via text or speech. | Mental health interventions ($n = 4$), Exercise coaching ($n = 3$) |
| Diagnostic support | 3 | Augment diagnostic ability of clinicians based on patients' presenting symptoms. | Triage ($n = 1$), Differential diagnosis generator ($n = 1$), Diagnosis of vestibular disorders ($n = 1$) |
| Prediction models | 3 | Use input data to determine future likelihood of certain events. | Prediction of undiagnosed AF ($n = 1$), Prediction of asthma exacerbation ($n = 1$), Patient language prediction for telephone calls ($n = 1$) |
| Automated drug dosage | 3 | Interpreting biological parameters and adjusting drug dose automatically. | Insulin ($n = 2$), Analgesia ($n = 1$) |
| Personalised lifestyle recommendations | 3 | Providing tailored lifestyle advice for patients with chronic conditions. | Heart failure ($n = 1$), Hypertension ($n = 1$), T2DM ($n = 1$) |
| Software interventions for patients | 3 | Patient education and therapeutic interventions delivered through software. | ADHD cognitive stimulation ($n = 1$), VR limb rehabilitation ($n = 1$), Interactive educational materials in T2DM ($n = 1$) |
| ECG classification | 2 | Automated diagnosis or interpretation based on ECG findings. | Identifying AF recurrence ($n = 1$), Detection of low ejection fraction ($n = 1$) |
| Personalised patient messaging | 2 | Attempting to increase effectiveness of patient reminders through personalisation. | Dentist recall visits ($n = 1$), Statin adherence ($n = 1$) |
| Prescription assistance | 2 | Integration with electronic prescription systems to improve prescribing safety. | Identification of high-risk prescriptions ($n = 1$), Reducing inappropriate antibiotic prescribing ($n = 1$) |
| Miscellaneous | 4 | Do not fit in other categories. | Nursing documentation assistance ($n = 1$), Augmented reality glasses ($n = 1$), Personalised patient decision aid ($n = 1$), Speech recognition ($n = 1$) |

*CT* Computed tomography, *ECG* Electrocardiogram, *AF* Atrial fibrillation, *ADHD* Attention deficit hyperactivity disorder, *VR* Virtual reality, *T2DM* Type 2 Diabetes Mellitus

human speech or text prompts and generate responses; specific uses within the included studies were digital mental health assistance and exercise coaching, used as a supplement to healthcare professional-guided therapy. All RCTs had two arms, with the exception of one study that investigated two different chatbot interventions against a control intervention simultaneously (delivering personalised exercise coaching by smart speaker or by text messaging)[25]. Full classification and description of interventions is shown in Table 1.

AI interventions were placed into categories according to level of human oversight: 'data presentation' ($n = 27$, 43%), 'clinical decision support' ($n = 14$, 22%), 'conditional automation' ($n = 6$, 10%) and 'high automation' ($n = 16$, 25%). No AI interventions were determined to have 'full automation'. More broadly, interventions were classified as assistive (non-autonomous) ($n = 41$, 63%) or autonomous ($n = 22$, 34%). Two studies (3%) did not report sufficient detail to determine level of human oversight.

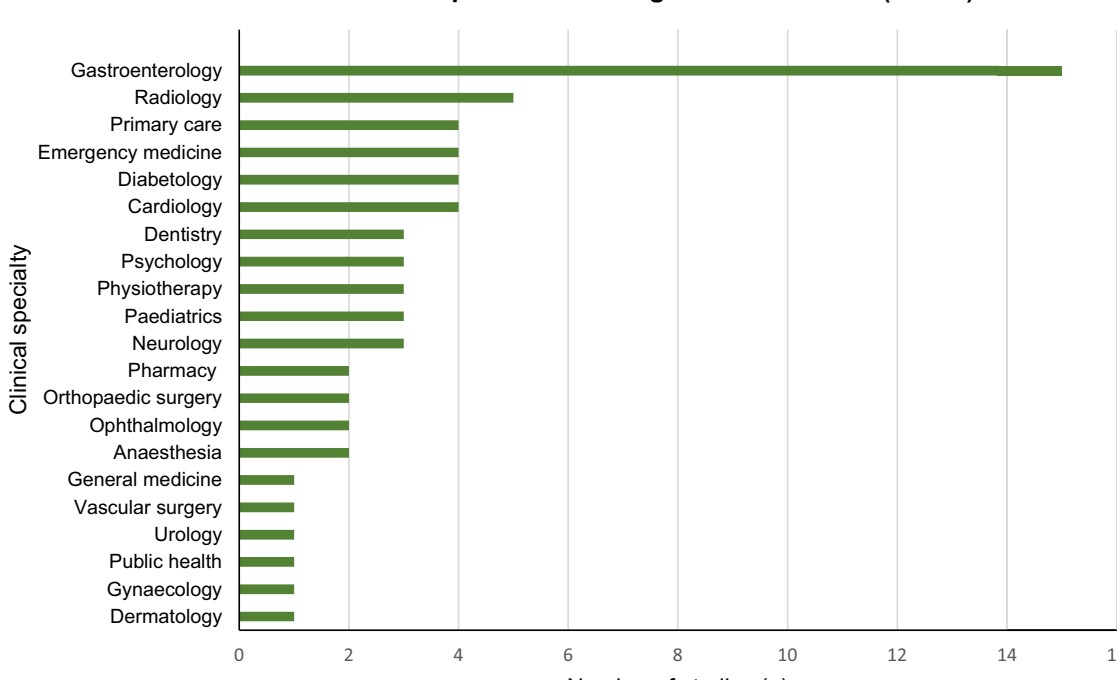

**Fig. 3 |** Distribution of clinical specialties amongst included RCTs, showing a high prevalence of interventions within gastroenterology.

**Table 2 | Overall CONSORT-AI concordance according to self-reported use of guidelines**

| Subgroup | Number | Overall CONSORT-AI concordance (%) median (IQR) |
|---|---|---|
| All included randomised controlled trials | 65 | 90 (77–94) |
| Those that reported use of CONSORT-AI | 10 | 96 (94–99) |
| Those that reported use of CONSORT 2010 | 9 | 92 (92–94) |
| Those that reported use of other CONSORT guidelines | 5 | 90 (81–94) |
| Those that did not report use of any guidelines | 41 | 84 (62–91) |

## Clinical specialty of interventions

When grouped by clinical specialty, most RCTs were in gastroenterology (*n* = 15, 23%), followed by radiology (*n* = 5, 8%), primary care (*n* = 4, 6%), emergency medicine (*n* = 4, 6%), diabetology (*n* = 4, 6%) and cardiology (*n* = 4, 6%). The full distribution of clinical specialties is shown in Fig. 3.

## Journal of publication

The 65 RCTs were published across 52 unique medical journals. As of May 2023, only two of the included journals (4%) explicitly mandated CONSORT-AI in their online submission guidelines (The Lancet Digital Health, The Lancet Gastroenterology) and one (2%) recommended CONSORT-AI without an explicit mandate (Ophthalmology Science). However, CONSORT 2010 was explicitly mandated by 28 journals (54%) and recommended without mandate in a further seven journals (13%). The EQUATOR Network (www.equator-network.org) is a comprehensive catalogue of reporting guidelines (including CONSORT-AI) and was recommended by 23 journals (44%) in total, of which eight (15%) specifically mandated its use to locate relevant reporting guidelines. Most journals that recommended use of the EQUATOR Network also explicitly recommended CONSORT 2010 (*n* = 21, 91%).

## Overall CONSORT-AI concordance

Overall median concordance to all CONSORT-AI items (comprising 14 AI-specific items and 37 non-AI-specific items) across all 65 included RCTs was 90% (IQR 77–94%). Two studies (3%) demonstrated 100% concordance[34,63]. Median overall CONSORT-AI concordance varied between geographical regions: China (86%, IQR 59–92%), USA (92%, IQR 90–94%), Japan (92%, IQR 86–96%) and Europe (93%, IQR 87–96%).

Ten RCTs (15%) explicitly reported use of CONSORT-AI, nine (14%) reported use of CONSORT 2010 only, five (8%) reported use of CONSORT-EHEALTH and 41 RCTs (63%) did not explicitly report use of any reporting guidelines. One study discussed CONSORT-AI in the limitations but did not make use of them, and instead reported according to CONSORT 2010[53]. Median overall CONSORT-AI concordance amongst studies that reported use of CONSORT-AI was 96% (IQR 94–99%), compared to 92% (IQR 92–94%) for those that used CONSORT 2010 only, 90% (IQR 81–94%) for those that used CONSORT-EHEALTH, and 84% (IQR 62–91%) for those that did not mention use of any reporting guidelines (see Table 2).

Given CONSORT 2010 has been widely adopted for many years, and the additional AI-specific items are relatively new recommendations, the next section will discuss reporting of AI-specific items and non-AI specific items separately.

## AI-specific CONSORT-AI items

When considering the 14 AI-specific CONSORT-AI items only, median concordance across all studies was 86% (IQR 71–93%). Just over half of studies (*n* = 36, 55%) reported 12 or more of the 14 checklist items, with four studies (6%) reporting 9 or fewer items. Of the six studies (9%) that

**Table 3 | Percentage concordance with AI-specific CONSORT-AI items**[8]

| CONSORT-AI checklist (AI-specific items) | Concordance (%)* |
|---|---|
| 1a,b (i) Indicate that the intervention involves artificial intelligence/machine learning in the title and/or abstract and specify the type of model. | 89% |
| 1a,b (ii) State the intended use of the AI intervention within the trial in the title and/or abstract. | 100% |
| 2a (i) Explain the intended use of the AI intervention in the context of the clinical pathway, including its purpose and its intended users (for example, healthcare professionals, patients, public). | 98% |
| 4a (i) State the inclusion and exclusion criteria at the level of participants. | 98% |
| 4a (ii) State the inclusion and exclusion criteria at the level of the input data. | 74% |
| 4b Describe how the AI intervention was integrated into the trial setting, including any onsite or offsite requirements. | 98% |
| 5 (i) State which version of the AI algorithm was used. | 20% |
| 5 (ii) Describe how the input data were acquired and selected for the AI intervention. | 97% |
| 5 (iii) Describe how poor quality or unavailable input data were assessed and handled. | 63% |
| 5 (iv) Specify whether there was human–AI interaction in the handling of the input data, and what level of expertise was required of users. | 97% |
| 5 (v) Specify the output of the AI intervention. | 100% |
| 5 (vi) Explain how the AI intervention's outputs contributed to decision-making or other elements of clinical practice. | 100% |
| 19 Describe results of any analysis of performance errors and how errors were identified, where applicable. If no such analysis was planned or done, justify why not. | 77% |
| 25 State whether and how the AI intervention and/or its code can be accessed, including any restrictions to access or re-use. | 42% |

*Concordance defined as proportion of "Yes" or "N/A" responses across all studies, rounded to nearest whole number.

**Table 4 | RCT CONSORT-AI concordance according to reporting guideline mandates from their journals of publication**

| Guidelines mandated by journal of publication | Number of RCTs | Concordance with all CONSORT-AI items (%) median (IQR) | Concordance with AI-specific items (%) median (IQR) | Concordance with non-AI-specific items (%) median (IQR) |
|---|---|---|---|---|
| CONSORT-AI | 2 | 100 (–) | 100 (–) | 100 (–) |
| CONSORT 2010 | 30 | 92 (90–95) | 86 (79–93) | 92 (90–95) |
| No guidelines mandated | 35 | 82 (61–90) | 79 (71–93) | 81 (57–92) |

achieved 100% concordance, five had reported use of the CONSORT-AI checklist and one had not. Median concordance varied between geographical regions: China (79%, IQR 71–85%), USA (86%, IQR 73–91%), Japan (86%, IQR 82–96%) and Europe (93%, IQR 86–93%).

Concordance also varied between AI-specific items (Table 3). Concordance was especially low for items 5 (i) (stating algorithm version) and 25 (whether the AI intervention / code can be accessed): 20% and 42%, respectively. Items 4a (ii) (inclusion criteria for input data), 5 (iii) (handling of poor-quality input data) and 19 (analysis of performance errors) were also relatively poorly reported. 100% concordance was observed for items 1a,b (ii) (stating intended use of intervention), 5 (v) (stating output of intervention) and 5 (vi) (explaining how the output contributed to decision-making).

There was no significant correlation between date of publication and CONSORT-AI concordance (Spearman's $r = -0.21$, $p = 0.091$). However, this exploratory analysis was limited by the small number of studies and narrow date range.

**Non-AI-specific CONSORT-AI items**
For the 37 non-AI-specific CONSORT-AI items (i.e., those contained within CONSORT 2010), median concordance across all RCTs was 92% (IQR 76–97%). Eight studies (12%) demonstrated 100% concordance with the non-AI-specific items, of which seven had explicitly reported use of CONSORT 2010 or CONSORT-AI. Median non-AI-specific CONSORT-AI concordance varied between geographical regions: China (88%, IQR 54–95%), USA (97%, IQR 93–97%), Japan (95%, IQR 85–99%) and Europe (95%, IQR 87–98%). Mean concordance for non-AI-specific items can be found in Supplementary Data 2.

There were several non-AI-specific CONSORT-AI items that were relatively poorly reported, including item 10 (who generated allocation sequence / enrolled participants / assigned participants to interventions) at 51%, and item 24 (access to full trial protocol) at 31%.

Reporting was also suboptimal around sample size calculation, randomisation methods, reporting harms / unintended effects and trial registration details (Supplementary Data 2).

**Journal reporting guideline mandates**
Overall, reporting concordance with CONSORT-AI was good regardless of whether journals mandated its use (Table 4). Median CONSORT-AI concordance was higher for RCTs published in journals where CONSORT-AI was mandated ($n = 2$, 3%), at 100%, versus 90% (IQR 76–94%) for RCTs published in journals that did not mandate CONSORT-AI ($n = 63$, 97%).

RCTs published in journals where CONSORT 2010 was mandated ($n = 30$, 46%) had a higher overall median CONSORT-AI concordance of 92% (IQR 90–95%), versus 82% (IQR 61–90%) where CONSORT 2010 was not mandated ($n = 35$, 54%). This is primarily attributable to non-AI-specific item concordance, which had a median of 92% (IQR 90–95%) versus 81% (IQR 57–92%) in CONSORT 2010 mandated versus non-mandated journals, respectively. AI-specific items also showed higher concordance when CONSORT 2010 was mandated, with median 86% (IQR 79–93%) versus 79% (IQR 71–93%).

## Discussion
The primary aim of this review was to determine the extent to which published RCTs report according to the CONSORT-AI extension since its publication in September 2020. We found 65 RCTs evaluating AI interventions in a variety of clinical settings and countries. Only 10 RCTs mentioned use of CONSORT-AI and 9 mentioned use of CONSORT 2010. Despite this, concordance with CONSORT-AI was generally high. There remains notable areas of poor reporting, such as stating the AI algorithm's version, explaining whether or how the AI algorithm could be accessed, and most studies did not report details and availability of the full study protocol. From a journal mandate point of view,

only 3 out of 52 journals instructed or recommended use of the CONSORT-AI checklist. It was unsurprising that journal mandates for use of CONSORT-AI were associated with greater concordance with CONSORT-AI reporting items (100% concordance versus 90%). However, we also found that AI RCTs published in journals endorsing CONSORT 2010 were more transparently reported compared to journals endorsing no reporting guidelines – according to CONSORT-AI specific considerations (92% concordance versus 82%). This may point towards a higher level of editorial scrutiny in journals which promote better reporting practices.

We found poor reporting for item 5 (i), regarding the statement of algorithm version used, at a median of only 20%. Lack of reporting on algorithm versioning (or other type of traceable identifier) raises significant concerns when appraising evidence of past and future studies of the same AI intervention. Without a traceable identifier, significant adjustments and updates (if any) that have been made over the lifetime of the AI intervention cannot be tracked and compared, so comparison between studies becomes difficult. This is becoming more relevant as AI medical devices are coming to market with referenced evaluation evidence published years ago. Stating whether the AI intervention or its code could be accessed (item 25) was also poorly reported, with median concordance of 40%. This may impede the ability of other researchers to achieve independent evaluation and potentially replication of findings, especially when the AI device is not a commercially available product and there is no named manufacturer. The remaining AI-specific CONSORT-AI items with lower concordance were item 4a (ii), regarding inclusion criteria at the level of the input data and item 5 (iii), regarding how poor-quality input data was handled – both important for reproducibility of the intervention in future trials and real-world use. Additionally, relatively few RCTs reported item 19, regarding results of performance error analysis, indicating the exploration of AI errors in an attempt to gain further insight into the nature and cause of AI failures, as well as their consequences, remains non-standard practice.

Overall, concordance with non-AI-specific CONSORT-AI items was higher than for AI-specific items, at 86% (IQR 71–93%) versus 92% (IQR 76–97%), likely due to its longstanding ubiquity amongst the medical scientific community and widespread acceptance as the standard of reporting. Despite this, low concordance was observed for several items, most notably providing access to the full trial protocol (item 24) with a concordance of only 31%. This has implications for reporting transparency as unreported protocol deviations may obscure bias in the methodology and presentation of findings.

Most RCTs did not mention using specific reporting guidelines and only 10 out of the 65 included studies explicitly reported use of CONSORT-AI. This low uptake may be explained by lack of journal mandates in instructions to authors. The CONSORT-AI extension was mandated by only two of the 52 journals in which the included studies were published, with one additional journal recommending its use without mandate. Other journals either recommended CONSORT 2010 or signposted to generic resources like the EQUATOR Network, where finding CONSORT-AI would be up to the individual authors' initiative.

Previous research on instructions for authors in high impact factor journals, in the context of CONSORT 2010, has shown that journal endorsement is sometimes lacking – especially in the endorsement of specific extensions[75]. Following the publication of CONSORT-AI in late 2020, the working group has reached out to editors of over 110 medical journals, raising awareness of the availability of these new standards. CONSORT-AI has been referenced by policy and regulatory bodies including the WHO[76], FDA[77] and MHRA[78], and has received over 400 citations to date. Despite this, we found that there remains low journal uptake, so mechanisms to lower the bar for adoption may require further consideration. One method to address this could be through editorial systems with tick boxes for authors to indicate the type of work being submitted, where the appropriate reporting checklist could be automatically delivered to be submitted with the paper. Such mechanisms will help ensure transparent reporting whilst reducing the burden on journal editors.

This systematic review provided an opportunity to assess the applicability and interpretation of CONSORT-AI recommendations across a diverse range of RCTs published since September 2020. Given the fast-moving nature of the field, this review also served as a mechanism for reflecting on clarity and applicability of the CONSORT-AI extension and to consider whether the items remain applicable to new and emerging types of AI interventions.

For item 1a,b (i) – "indicate that the intervention involves artificial intelligence/machine learning in the title and/or abstract and specify the type of model" – the type of AI model was frequently not specified within the abstract. A decision was made in this review to not impose stringent requirements for the "type of model" component. It is debatable how meaningful a short description of model type in the title and abstract can be and perhaps a full description of the AI model is more relevant for diagnostic test accuracy studies model development and validation studies (where STARD-AI and TRIPOD + AI are more relevant reporting guidance, respectively)[79,80].

We also want to reflect on difficulties experienced by our reviewers when assessing certain items, which may be due to poor reporting by authors of the RCTs, but could also indicate a lack of clarity in the item itself. For example, for item 5 (iii) – "describe how poor-quality or unavailable input data were assessed and handled", it was difficult to interpret the information provided as a separate consideration from item 4a (ii) – "state the inclusion and exclusion criteria at the level of the input data". There were several disagreements during data extraction which required discussion, as it was unclear whether some RCTs were discussing input data inclusion and exclusion criteria (item 4a (ii)) or the quality of the actual input data post-inclusion (item 5 (iii)). Further elaboration may be needed to differentiate these two criteria in the CONSORT-AI documentation and/or provide authors with more specific reporting instructions. This item was also difficult to apply to certain AI interventions, especially AI-assisted endoscopy. Some AI interventions will, by design, automatically exclude data that cannot be processed, which is desirable from a safety perspective. This means that item 5 (iii) may be inapplicable and be less likely to be reported as a result.

Similarly, for item 19 – "describe results of any analysis of performance errors and how errors were identified" – assessment of concordance was challenging for certain AI interventions, especially those involving digital therapeutics (for example, AI-delivered cognitive behavioural therapy, counselling or rehabilitation). Analysis of performance errors was rarely performed in these studies, but we also found it difficult to define what performance errors could look like and how they could be measured within a trial setting for such interventions. Errors could be subtle and difficult to verify beyond obviously nonsensical responses. It may be appropriate to report an evaluation of harmful effects caused by the AI intervention, including disparate harms across subpopulations, however these effects may be difficult to detect. As applications of AI technologies evolve, it is important that guidelines maintain relevance. Given the rapid growth of digital therapeutics and medical large language models, this could be an area of focus for subsequent CONSORT-AI iterations[81,82].

An additional reflection is that this review identified a high proportion of trials evaluating AI-assisted endoscopy interventions for gastroenterology. This is in keeping with findings from a recent review by ref. 83, and may be explained by the challenge of assessing the performance of these devices in non-interventional trials (given AI-assisted endoscopy is implemented in real-time). For other AI interventions such as image classification systems, observational retrospective or prospective studies can provide indications of diagnostic accuracy, with evaluation of the downstream impact to health and

resource outcomes less commonly evaluated. Furthermore, the performance of AI-assisted endoscopy is typically evaluated by measuring adenoma detection rate as an outcome. This can only be determined by performing polyp removal and confirmation using histopathology; therefore, interventional trials are necessary.

Previous systematic reviews have used the CONSORT-AI checklist to evaluate reporting completeness of RCTs involving AI interventions in healthcare[84–86]. However, these differ from the current systematic review in terms of methodology (for example, using a less sensitive search strategy consisting of three search terms[85]) and incomplete application of CONSORT-AI (specifically, excluding three of the 14 AI-specific CONSORT-AI items[84]). Additionally, these reviews executed their literature searches in 2021 or earlier, less than a year after CONSORT-AI was published. Our systematic review used a robust search strategy, including clinical trials registries, and was carried out in conjunction with CONSORT-AI authors to ensure that each item was interpreted correctly. Furthermore, this review covers a two-year article submission period following publication of CONSORT-AI to provide a fairer assessment of initial uptake.

One limitation of this systematic review was the potential for incomplete study retrieval despite best efforts to maximise sensitivity. For example, some RCTs published in computer science journals did not explicitly identify as RCTs in the title, abstract or keywords, which could mean other similar trials were not retrieved by the literature search. Furthermore, indexing errors for study status in trial registry entries may have led to incorrect exclusion of published studies that had not been updated in the clinical trial registry. However, an attempt was made to mitigate this by searching relevant trial registration numbers through Google Search if no linked publication was included on the trial registry page. It should be acknowledged that publications included in our review may have been submitted soon after publication of the CONSORT-AI guidelines (September 2020) and may not have had sufficient time to be drafted in accordance due to the length of editorial processes. Our search strategy includes terms describing AI and ML, which inevitably confounds concordance with CONSORT-AI item 1a,b (i): "Indicate that the intervention involves artificial intelligence / machine learning in the title and/or abstract and specify the type of model." However, this is a necessary keyword for literature searching and therefore an unavoidable confounder. Finally, non-English language RCTs were excluded, which has the potential to introduce bias, particularly when considering the diverse geographical spread of RCTs.

In conclusion, the results of this systematic review have shown that in the 2-year period since publication of CONSORT-AI in September 2020, most AI-specific CONSORT-AI items were well-reported across relevant studies. However, a small number of specific items remain poorly reported. As with other reporting guidelines, the potential value of CONSORT-AI in improving reporting would be further enhanced by encouraging adoption, for example, through recommendations (or even mandates) from journals or funders. This systematic review has indirectly served as a test of the feasibility and usability of CONSORT-AI, indicating that some minor modifications in future updates to the checklist may help improve accessibility to authors and maintain relevance to the latest AI technologies. Arguably it is still early days to evaluate the impact of CONSORT-AI, given that many RCTs take years to complete and become published. Future reviews of AI RCTs could also compare these findings to new and ongoing RCTs that will be published in the coming years.

## Methods

This systematic review is reported according to the PRISMA 2020 statement[74]. The protocol was prospectively registered on the Centre for Open Science's Open Science Framework (OSF) Registry (doi.org/10.17605/OSF.IO/CRF3Q).

### Search strategy

A combination of keywords and MeSH terms was used to identify RCTs on interventions involving AI, for example: "artificial intelligence", "decision support system", "deep learning" and "neural network", in addition to specific terms such as "naïve bayes", "random forest" and "multilayer perceptron". A modified version of the Cochrane RCT sensitivity and precision maximising filter was used to improve relevant article retrieval[87]. The search strategy was developed in conjunction with an information specialist and was not adapted from any previous reviews. Keywords and subject headings were adjusted for each database as required. Database search strategies and PRISMA-S checklist are included in Supplementary Information.

MEDLINE, Embase and Cochrane Central databases were searched on 19th September 2022. Clinical trial registries, including the International Clinical Trials Registry Platform and ClinicalTrials.gov, were searched for completed studies on the same date. Articles published from 9th September 2020 onwards were retrieved for screening, following the date of CONSORT-AI publication. Articles were restricted to English language. Reference lists of included articles and identified secondary research sources were screened for relevant articles before exclusion. The database searches were not repeated.

### Study selection

Eligibility criteria were primary reports of RCTs involving AI interventions within any healthcare setting, available in the English language. AI interventions were defined as any devices or tools with an AI or machine learning component, determined by reviewers during screening. Conference abstracts, protocols and studies primarily evaluating robotics were excluded. Articles submitted to the journal of publication prior to the release of CONSORT-AI guidelines (September 2020), determined by online publication history, were excluded.

Covidence systematic review software (2022) was used to collate references, deduplicate and screen for inclusion at both title / abstract and full-text stages[88]. Title and abstracts were independently screened by two authors (AM and VN). Full-text articles of eligible studies were retrieved and independently assessed in detail by two authors (AM and VN) before inclusion or exclusion, with reasons given for the final decision. Disagreements were resolved by discussion or by a senior author (XL).

### Data extraction

Two authors (AM and XL) independently extracted data from the final selection of RCTs, including study characteristics (first author, date of publication, country of study, medical specialty, publishing journal, number of study sites, blinding, study duration, sample size, randomisation technique, experimental and control interventions, AI characteristics, level of human oversight, use of CONSORT-AI) and concordance with the 14 AI-specific items of the CONSORT-AI checklist. Level of human oversight was classified according to a graded autonomy model described by Bitterman et al., which included the categories: 'data presentation' (AI highlights areas for review by the clinician), 'clinical decision support' (AI calculates a risk score that is interpreted by the clinician), 'conditional automation' (AI acts autonomously with clinician as backup), 'high automation' (AI acts autonomously with no clinician backup or validation) and 'full automation' (as for 'high automation' but can be used across all populations or systems)[89]. Any conflicts were resolved by discussion. For each journal of publication of the included RCTs, online submission guidelines were accessed to determine the recommended RCT reporting guidelines, including whether CONSORT-AI was recommended or mandated. Journal submission guidelines and concordance with the 37 non-AI-specific items of CONSORT-AI were assessed by two authors (AM and BN), with any conflicts resolved by a senior author (XL). Risk of bias assessment was not conducted as this review was primarily concerned with completeness of reporting

for AI-specific considerations, rather than RCT outcomes and intervention effectiveness.

## Data synthesis

Primary analysis of CONSORT-AI concordance was assessed through percentage of RCTs reporting each item. Results relating to concordance are reported for all CONSORT-AI items, as well as AI-specific and non-AI-specific items separately. Concordance is then reported according to the country of RCT conduct to examine variations in reporting practice across geographies. Lastly, concordance is reported according to whether the journal of publication mandated or endorsed the use of CONSORT-AI, CONSORT 2010 and/or any other reporting guidance. Concordance was defined as fulfilment of all components of each CONSORT-AI item, or the item being non-applicable. This rule was applied to all items with the exception of item 1a,b (i) – "indicate that the intervention involves artificial intelligence/machine learning in the title and/or abstract and specify the type of model". After reviewing a sample of studies, we found the type of AI model was frequently not specified within the abstract. For this review, RCTs were considered to achieve this criterion as long as AI or ML were described, however stringent requirements for the "type of model" component were not applied. Analysis of study characteristics was performed using descriptive statistics and figures. An exploratory analysis that was not part of the original protocol was carried out using Spearman's Rank-Order Correlation to determine whether CONSORT-AI concordance had changed with later dates of publication. P-values under 0.05 were considered significant. Statistical analysis was performed using Statistical Package for Social Sciences (SPSS) for Windows, Version 25.0.

## Reporting summary

Further information on research design is available in the Nature Portfolio Reporting Summary linked to this article.

# Data availability

All data used in this study is referenced and publicly available. Supplementary information has been provided. Any further study materials, including data collection forms and data extracted from included studies, are available upon request to the corresponding author.

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

## Acknowledgements

We would like to thank Michael D. Howell for his role in reviewing our manuscript and Viknesh Sounderajah for his input in the conception of this review.

## Author contributions

AM: methodology, formal analysis, investigation, data curation, writing (original draft, review & editing); CDL, ROdV: supervision, writing (original draft, review & editing); BN and VN: investigation, writing (review & editing); AK: methodology, writing (review & editing); LFR, RG, GSC, DM, MM, LOR, SCR, MC, CK, CSL, CY, AWC, PK, and AB: writing (review & editing); AD: supervision, writing (review & editing); XL: conceptualisation, supervision, investigation, writing (original draft, review & editing).

## Competing interests

Several authors (X.L., D.M., A.D., C.K., L.F.R., C.L., A.W.C., M.C., P.K., G.S.C., R.G., L.O.R., M.M., C.Y., S.C.R., A.B.) were involved in the development of CONSORT-AI. MC receives funding from the NIHR, UK Research and Innovation (UKRI), NIHR BRC, the NIHR Surgical Reconstruction and Microbiology Research Centre, NIHR ARC West Midlands, NIHR Birmingham-Oxford Blood and Transplant Research Unit (BTRU) in Precision Transplant and Cellular Therapeutics, UKSPINE, European Regional Development Fund – Demand Hub and Health Data Research UK at the University of Birmingham and University Hospitals Birmingham

NHS Foundation Trust, Innovate UK (part of UKRI), Macmillan Cancer Support, UCB Pharma, GSK and Gilead. M.C. has received personal fees from Astellas, Aparito Ltd, CIS Oncology, Takeda, Merck, Daiichi Sankyo, Glaukos, GSK and the Patient-Centred Outcomes Research Institute (PCORI) outside the submitted work. X.L. and A.D. have received funding from the NHS AI Lab, The Health Foundation, NIHR, NIHR BRC, MHRA and NICE, outside the submitted work. A.D. and M.C. are supported by the NIHR Birmingham Biomedical Research Centre. The views expressed are those of the author(s) and not necessarily those of the NIHR or the Department of Health and Social Care. C.K. is an employee of Google, UK. L.F.R. is an employee of York Health Economics Consortium (YHEC). The remaining authors declare no competing interests.

## Additional information

[1]Brighton and Sussex Medical School, Brighton, UK. [2]Department of Primary Care and Public Health, Brighton and Sussex Medical School, Brighton, UK. [3]Birmingham and Midland Eye Centre, Sandwell and West Birmingham NHS Trust, Birmingham, UK. [4]Christ Church, University of Oxford, Oxford, UK. [5]University College London Medical School, London, UK. [6]Institute of Inflammation and Ageing, University of Birmingham, Birmingham, UK. [7]University Hospitals Birmingham NHS Foundation Trust, Birmingham, UK. [8]National Institute for Health and Care Research (NIHR) Birmingham Biomedical Research Centre, University of Birmingham, Birmingham, UK. [9]York Health Economics Consortium, University of York, Birmingham, UK. [10]Northwestern University Feinberg School of Medicine, Chicago, Illinois, USA. [11]Centre for Statistics in Medicine//UK EQUATOR Centre, Nuffield Department of Orthopaedics, Rheumatology and Musculoskeletal Sciences, University of Oxford, Oxford, UK. [12]Centre for Journalology, Clinical Epidemiology Program, Ottawa Hospital Research Institute, Ottowa, Canada. [13]Department of Bioethics, The Hospital for Sick Children, Toronto, Canada. [14]Genetics & Genome Biology Research Program, Peter Gilgan Centre for Research & Learning, Toronto, Canada. [15]Division of Clinical and Public Health, Dalla Lana School of Public Health, Toronto, Canada. [16]Australian Institute for Machine Learning, University of Adelaide, Adelaide, Australia. [17]Birmingham Health Partners Centre for Regulatory Science and Innovation, University of Birmingham, Birmingham, UK. [18]Centre for Patient Reported Outcomes Research (CPROR), Institute of Applied Health Research, College of Medical and Dental Sciences, University of Birmingham, Birmingham, UK. [19]NIHR Applied Research Collaboration (ARC) West Midlands, University of Birmingham, Birmingham, UK. [20]NIHR Blood and Transplant Research Unit (BTRU) in Precision Transplant and Cellular Therapeutics, University of Birmingham, Birmingham, UK. [21]Google Health, London, UK. [22]University of Washington, Seattle, WA, USA. [23]Nuffield Department of Women's and Reproductive Health, University of Oxford, Oxford, UK. [24]Health Data Research UK, London, UK. [25]Department of Medicine, Women's College Hospital. University of Toronto, Toronto, Canada. [26]NIHR Biomedical Research Centre at Moorfields, Moorfields Eye Hospital NHS Foundation Trust and UCL Institute of Ophthalmology, London, UK. [27]Department of Epidemiology, Harvard. T.H. Chan School of Public Health, Boston, MA, USA. [28]Department of Biomedical Informatics, Harvard Medical School, Boston, MA, USA. ✉e-mail: xliuphone@gmail.com

