## [Peer Review File · Nature Communications]

Concordance of Randomised Controlled Trials for Artificial Intelligence Interventions with the CONSORT-AI Reporting GuidelinesREVIEWER COMMENTS

Reviewer #1 (Remarks to the Author):

The authors evaluate the performance of the AI+medicine community in following guidelines set by the CONSORT-AI working group regarding trials of AI. Notably, there are few AI RCTs, so there is a balance between encouraging more trials and designing trials with the most rigor possible. (Ideally we would have both, but an open question whether more trials of lower quality is preferable to no AI RCTs. I personally slightly lean towards that, so feel less that it is a bad outcome all trials dont follow CONSORT-AI guidelines and imagine could be similarly true for all RCTs and CONSORT in general.) It seems favorable that 90% concordance with the reporting guidelines.

As such, while I agree with the guidelines, I think the last statement is maybe too strong "Despite generally high concordance amongst recent AI RCTs, adoption of CONSORT-AI remains low. Encouraging journal and funder mandates may enable the potential value of CONSORT-AI to be realized." If the recommendations are mostly followed (90%), why should be enforce its explicit recommendation? Many general clinical trials do not explicitly say it follows CONSORT, even if a CONSORT checklist is filled out.

This is a good work and worth highlighting areas of shortcoming. I would like to have seen a breakdown for commercial vs academic projects with regard to concordance with CONSORT-AI. Perhaps version control is stronger in commercial products while code sharing is frequent in academic projects?

Reviewer #2 (Remarks to the Author):

In terms of the systematic review methodology this study was well-conducted. I have only two minor comments:

Please clarify whether or not the full-text screening was done independently.

Please add N=267 to the top of the Articles excluded box in the PRISMA diagram (Fig 1).

Reviewer #1 (Remarks to the Author):

The authors evaluate the performance of the AI+medicine community in following guidelines set by the CONSORT-AI working group regarding trials of AI. Notably, there are few AI RCTs, so there is a balance between encouraging more trials and designing trials with the most rigor possible. (Ideally we would have both, but an open question whether more trials of lower quality is preferable to no AI RCTs. I personally slightly lean towards that, so feel less that it is a bad outcome all trials dont follow CONSORT-AI guidelines and imagine could be similarly true for all RCTs and CONSORT in general.) It seems favorable that 90% concordance with the reporting guidelines.

As such, while I agree with the guidelines, I think the last statement is maybe too strong "Despite generally high concordance amongst recent AI RCTs, adoption of CONSORT-AI remains low. Encouraging journal and funder mandates may enable the potential value of CONSORT-AI to be realized." If the recommendations are mostly followed (90%), why should be enforce its explicit recommendation? Many general clinical trials do not explicitly say it follows CONSORT, even if a CONSORT checklist is filled out.

We have updated the abstract to clarify the rationale behind this recommendation.

This is a good work and worth highlighting areas of shortcoming. I would like to have seen a breakdown for commercial vs academic projects with regard to concordance with CONSORT-AI. Perhaps version control is stronger in commercial products while code sharing is frequent in academic projects?

AI interventions were developed by academic institutions (n = 28, 43%), commercial organisations (n =25, 38%) or a combination of both (n = 4, 6%). 8 trials (12%) did not report this information. Despite this, it was not possible to consistently determine whether RCTs were commercially led or academic in nature. Analysis of the RCTs where this data was available showed no clear differences, however we feel this analysis is too limited to show.

Reviewer #2 (Remarks to the Author):

In terms of the systematic review methodology this study was well-conducted. I have only two minor comments:

Please clarify whether or not the full-text screening was done independently.

This was done independently. Clarified on page 6.

Please add N=267 to the top of the Articles excluded box in the PRISMA diagram (Fig 1).

Figure 1 updated.

REVIEWERS' COMMENTS

Reviewer #1 (Remarks to the Author):

The authors answer my questions and seem appropriate.

Reviewer #2 (Remarks to the Author):

Happy with the minor amendments made.